# Effect of Shape, Number, and Location of Openings on Punching Shear Capacity of Flat Slabs



**Ekkachai Yooprasertchai [1]**, **Yonlada Tiawilai [1]**, **Theerawee Wittayawanitchai [1]**, **Jiranuwat Angsumalee [1]**, **Panuwat Joyklad [2,\*]** and **Qudeer Hussain [3]**

[1] Construction Innovations and Future Infrastructure Research Center (CIFIR), Department of Civil Engineering, Faculty of Engineering, King Mongkut's University of Technology Thonburi, Bangkok 10140, Thailand; ekkachai.yoo@kmutt.ac.th (E.Y.); yonlada.tiawilai1@gmail.com (Y.T.); dome84898@gmail.com (T.W.); Jiranuwat27@gmail.com (J.A.)

[2] Department of Civil and Environmental Engineering, Srinakharinwirot University, Nakhon Nayok 26120, Thailand

[3] Center of Excellence in Earthquake Engineering and Vibration, Department of Civil Engineering, Chulalongkorn University, Bangkok 10330, Thailand; ebbadat@hotmail.com

\* Correspondence: panuwatj@g.swu.ac.th

**Abstract:** Experimental evidence have proved that punching shear capacity of flat slabs deteriorate with the presence of openings located within the critical perimeter around columns. It is understood that this deterioration varies inversely with the distance of openings from column's face. However, effect of the shape of openings on punching shear capacity is not well known. This study presents experimental results of 14 flat specimens to investigate the effects of the number (2 and 4), shape (circular, square, and rectangular), and location (1 and 4 times of slab's thickness from column's face) of openings on punching shear strength. It was found that circular openings had least influence on punching capacity followed by square and rectangular openings, respectively. Further, placing openings at a distance of four times the slab's thickness from column's face had minimal impact on punching capacity. Further, increasing the number of openings from 2 to 4 substantially reduced the punching capacity. An effort was made to predict the punching capacities of all specimens using the descriptive equations of ACI 318-19 and Eurocode 2. Mean of the ratio of experimental to analytical results and standard deviation of ACI equations were found to be more accurate than those of Eurocode 2 predictions.

**Keywords:** punching shear; effect of shape; ACI 318-19; Eurocode 2; flat slabs





## 1. Introduction

Nowadays, flat slabs are widely used because of their simple reinforcement details, reduced building heights, rapid construction etc. A flat slab without beams is less rigid and open to larger vertical displacements than conventional slabs [1,2]. Flat slabs are prone to punching shear due to the accumulation of high shear stresses in the vicinity of slab-column junction [3,4]. Punching shear failures are undesirable for their progressive occurrence without any signs of warnings [5]. Openings in flat slabs are usually provided to accommodate plumbing, electric wires, air-conditioning systems, etc. Experimental studies have demonstrated that the presence of such openings within the critical punching shear perimeter around the periphery of slab-column interface can substantially reduce the shear capacity of flat slabs.

El-Salakawy et al. [6] reported the results of on the effect of shear studs on the behavior of slab-column edge connections with openings. They concluded that the existence of openings substantially reduced the stiffness of slab-column edge connection. They further recommended that openings as large as the size of the column should not be constructed as shear studs had minimal effect in this situation. Teng et al. [7] focused on flat plate slabs

with openings on rectangular columns. They tested 20 slab specimens under concentrated loads and found that openings punching strengths of flat plate slabs were substantially reduced in the presence of openings. They further recommended that if the placement of an opening becomes inevitable, the best place is to construct along the longer side of the column. Borges et al. [8] carried out punching shear tests on 13 flat plate slabs with/without openings or/and shear reinforcement. Openings (1 or 2) were constructed along the short side of the column. It was shown that for the relatively small openings studied, the provision of continuous bars adjacent to openings—to replace the areas of reinforcement—seemed to be an acceptable approach to flexural design. Anil et al. [9] carried out an experimental investigation on the proximity (adjacent o column and at 300 mm away), location (parallel or diagonal to column), and size (300 mm × 300 mm and 500 mm × 500 mm) of openings on punching capacities of two-way square slabs (2000 mm × 2000 mm × 120 mm) subjected to concentrated axial load applied from top of column (200 mm × 200 mm). It was reported that punching resistance decreased as the opening size increased. Further, a notable decrease in punching capacity was observed for the openings placed adjacent to column in comparison to openings placed 300 mm away from the column. Ha et al. [10] performed experimental investigation on 8 flat slabs considering the layout and number of openings. It was reported that an L-shaped opening layout around the corner of a column resulted in further reduction in addition to the loss of effective critical sections due to the existence of openings. Balomenos et al. [11] carried out deterministic parametric investigation on the effect of opening distance from column's face and found out that irrespective of the size of opening, opening placed at 4 h (where "h" is slab's thickness") from column's face does not have significant effect on punching capacity of flat slabs. Liberati et al. [12] tested 12 reinforced concrete slabs without shear reinforcement under symmetrical loading. Slabs were tested in 3 groups depending upon the number of openings adjacent to column. It was reported that energy dissipation capacity of slabs decreased as the number of openings increased. Slab with openings underwent larger displacements than slabs without openings.

Experimental studies have highlighted the adverse effects of openings on shear capacity of slabs. It was reported [13] that openings next to column reduce the concrete area that sustains punching shear. Shear stresses were increased due to the presence of openings, load conditions, and unbalanced moments generated from slab geometry. Further, it was shown that openings located at distances greater than four times of slab effective depth had negligible effect on slab's shear capacity and such slabs behaved similar those without any openings. Moreover, shear strength of slabs with very small openings (50 mm and 70 mm in diameter) located at distance twice the effective depth of slab from the abutment also remained unchanged. Another study [14] investigated the number and size of circular openings around column's periphery. Number of openings were varied from 2 to 4 while their size was varied as 75 mm, 100 mm, and 150 mm. It was concluded that slabs with two openings (100 mm and 150 mm) lost stiffness compared to the slab with four 75 mm openings. Further, size of openings influenced the rotational capacity of slabs.

Size and proximity of an opening from column's face contribute mainly against punching shear strength. It has been reported that both the size and distance of an opening from column's face adversely affect punching capacity [9,15]. Five square flat slabs (1800 mm × 1800 mm × 130 mm) supported on square columns (150 mm × 150 mm) with square openings (150 mm × 150 mm) at different distances from column's face were tested until failure. Results revealed the presence of very high shear stresses concentrated in the regions between column and the opening. It was further concluded that any opening located farther than three times of slab's effective depth from column's face had no effect on the shear capacity of slab [16].

To the author's knowledge, no experimental work has been conducted till now to investigate the effect of opening shapes in flat slabs. However, an experimental study [17] was conducted to study the effect of the shape of opening on shear capacity of conventional slabs. A total of 15 slabs were tested with rectangular (200 mm × 150 mm and 150 mm ×

100 mm) and circular (150 mm and 100 mm) openings placed at the center of slab. It was found that the size of opening had a direct effect on the shear capacity of slabs. Further, the shear degradation caused by circular openings was lower than that caused by rectangular openings. The program involved only the placement of openings at the center of slab and testing under a Universal Testing Machine (UTM). Such testing methodology is far from replicating the actual loading scenario in flat slabs. Therefore, a more rational and practical approach is required to replicate the loading mechanism sustained by flat slabs in actual structures. Further, ACI 318-19 [18] and Eurocode 2 [19] impose limitations on the distance of an opening from column's face. ACI 318-19 and Eurocode 2 consider the maximum location at which an opening can still affect the punching shear capacity of flat slabs to be 4H and 6d from column's face, respectively (where "H" and "d" are slab thickness and effective depth, respectively). It is to be mentioned that ACI 318-19 recently adopted this distance to be 4H. Previous edition i.e., ACI 318-14 considered this distance to be 10H from column's face. However, information regarding the shape of an opening is yet to be established. Therefore, given the excessive use of flat slabs in modern-day construction industry along with the inevitable presence of openings in them, there is a significant need to further extend this domain to various opening shapes and their configurations.

The objectives of this study are to investigate the effect of the shape (circular and square) on the punching shear capacity of flat slab, to study the effects of the distance of an opening from the column's face, and to assess the applicability of existing punching shear capacity analytical formulations of ACI 318-19 [18] and Eurocode 2 [19] on the results of this study.

## 2. Experimental Program

### 2.1. Test Matrix

A total of 14 flat slabs were tested in this study. Each specimen was sized to 1000 mm, 1000 mm, and 80 mm in length, width, and depth, respectively. A single 100 mm × 100 mm steel plate was installed concentrically in each slab to simulate a column. Either 2 or 4 openings were provided in each specimen at opposite sides of the column. Size, shape, and location from column's face of openings in each specimen are provided in Table 1. Square (70 mm × 70 mm), rectangular (70 mm × 140 mm), and circular (70 mm) openings were used to investigate the shape effect of openings on punching shear strength. Effect of the proximity of an opening from column's face was studied by placing openings at 1 and 4 h from the column's face, where h is the slab's thickness i.e., 80 mm. Specimens were mainly divided in 2 groups depending upon the concrete strength. Distance at 4 h was chosen to cover a wide range of distance from column's face within which an opening could affect the punching shear capacity. It basically creates the lower bound as most recently ACI 318-19 has adopted this distance as the safe distance at which an opening could be placed with minimal effect on the punching capacity of slabs. The other distance (i.e., 1 h) was chosen for practical reasons and is expected to create the upper bound of the deterioration in punching capacity caused by the openings. Therefore, both distances were selected to cover the widest range possible. Each specimen had 1 specimen without any openings to serve as the reference/control specimen. The remaining 6 specimens were subdivided into 3 groups depending upon the shape of opening. Each subgroup of 2 specimens had similar opening shape but its distance varied to the above specified values. Nomenclature for specimens was chosen to represent the number, shape, and location of openings in the same order. For example, specimen 2S1H means that this specimen had 2 square openings located at 1 h from the column's face on each side of the column. Typical layout of the openings in each slab is presented in Figures 1 and 2 for group 1 and 2, respectively.

**Table 1.** Test Matrix.

| Specimen No. | Specimen Name | Compressive Strength of Concrete (MPa) | Number of Opening | Opening Shape | Opening Size (mm) | Location (mm) |
|---|---|---|---|---|---|---|
| 1 | CON1 | 20.18 | - | Control | Control | Control |
| 2 | 2S1H | 20.18 | 2 | Square | $70 \times 70$ | 80 (1 h) |
| 3 | 2C1H | 20.18 | 2 | Circular | Diameter 70 | 80 (1 h) |
| 4 | 2R1H | 20.18 | 2 | Rectangular | $70 \times 140$ | 80 (1 h) |
| 5 | 2S4H | 20.18 | 2 | Square | $70 \times 70$ | 320 (4 h) |
| 6 | 2C4H | 20.18 | 2 | Circular | Diameter 70 | 320 (4 h) |
| 7 | 2R4H | 20.18 | 2 | Rectangular | $70 \times 140$ | 320 (4 h) |
| 8 | CON2 | 29.71 | - | Control | Control | Control |
| 9 | 4S1H | 29.71 | 4 | Square | $70 \times 70$ | 80 (1 h) |
| 10 | 4C1H | 29.71 | 4 | Circular | Diameter 70 | 80 (1 h) |
| 11 | 4R1H | 29.71 | 4 | Rectangular | $70 \times 140$ | 80 (1 h) |
| 12 | 4S4H | 29.71 | 4 | Square | $70 \times 70$ | 320 (4 h) |
| 13 | 4C4H | 29.71 | 4 | Circular | Diameter 70 | 320 (4 h) |
| 14 | 4R4H | 29.71 | 4 | Rectangular | $70 \times 140$ | 320 (4 h) |

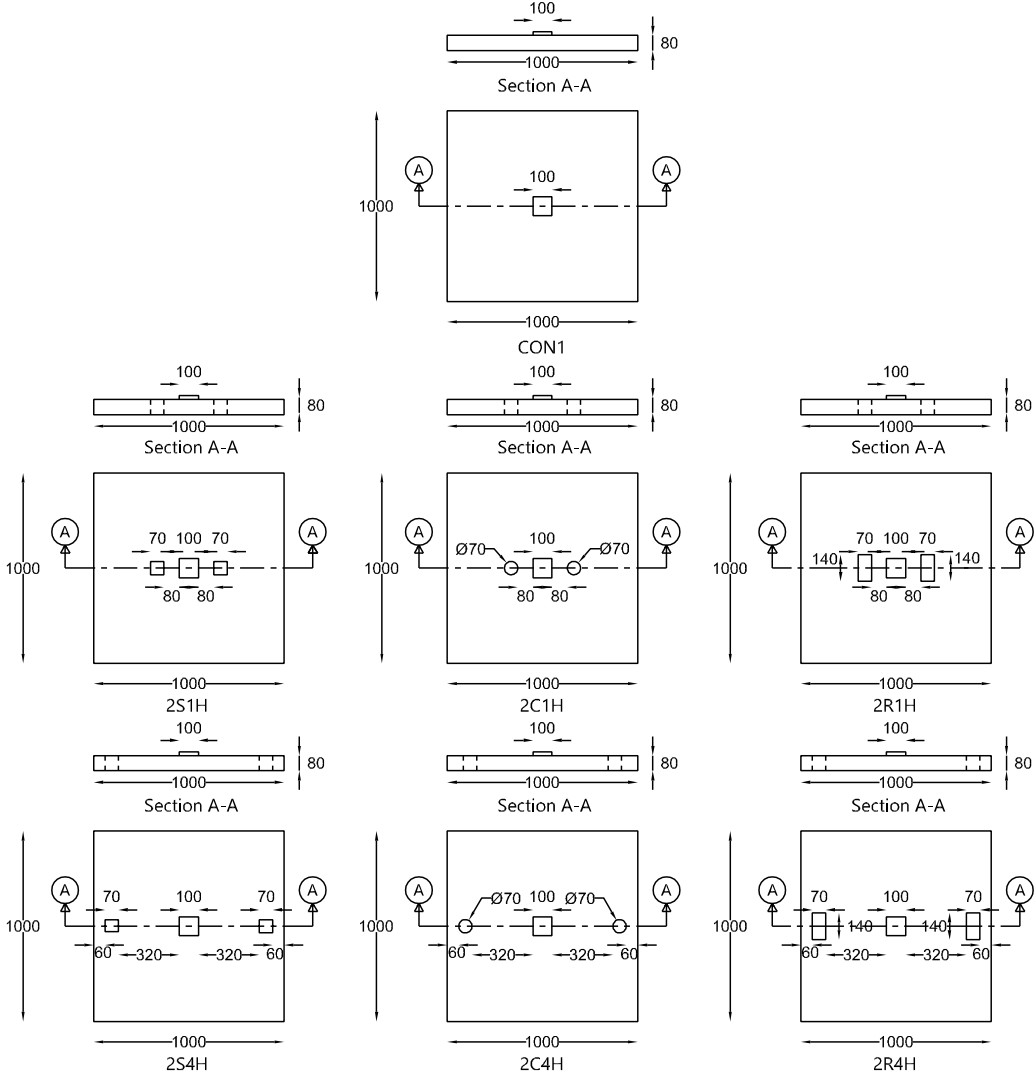

**Figure 1.** Layout of specimens with 2 openings (units: mm).

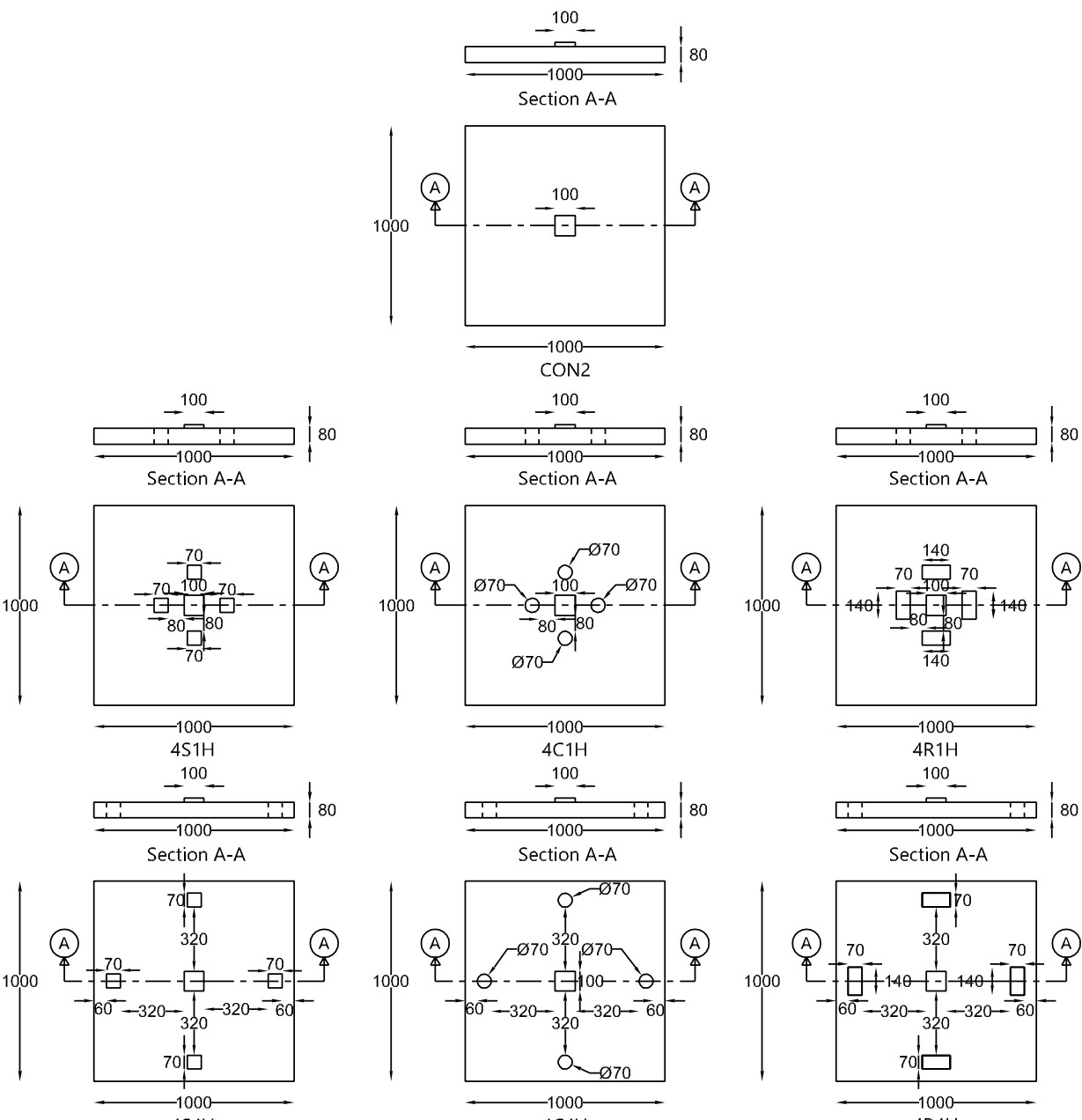

**Figure 2.** Layout of specimens with 4 openings (units: mm).

### 2.2. Specimen Design

All specimens were designed to be shear controlled. All specimens were provided with sufficient, and similar flexural reinforcement ratio (0.009) to have higher flexural strength than corresponding shear strengths. Typical layout of flexural reinforcement in each specimen is shown in Figure 3. ACI 318-19 [18] provisions were used to calculate the shear and flexural strengths of specimens. Shear strength provided by concrete is calculated using Equation (1).

$$v_C = \min \left[ \begin{array}{c} \left(0.33\lambda_s\lambda\sqrt{f_c'}\right), \left(\left(0.17 + \frac{0.33}{\beta}\right)\lambda_s\lambda\sqrt{f_c'}\right), \\ \left(\left(0.17 + \frac{0.83\alpha_s d}{b_0}\right)\lambda_s\lambda\sqrt{f_c'}\right) \end{array} \right] \tag{1}$$

where,

- $f'_c$ = concrete cylinder compressive strength (MPa)
- $d$ = effective slab thickness for shear (mm)
- $b_0$ = perimeter of shear critical section 0.5d from loading area periphery (mm)
- $\alpha_s$ = 40 for interior columns, 30 for edge columns, and 20 for corner columns.
- $\beta$ = The ratio of long to short sides of the columns, concentrated load, or reaction area.
- $\lambda_s = \sqrt{\dfrac{2}{1+0.004d}} \leq 1$

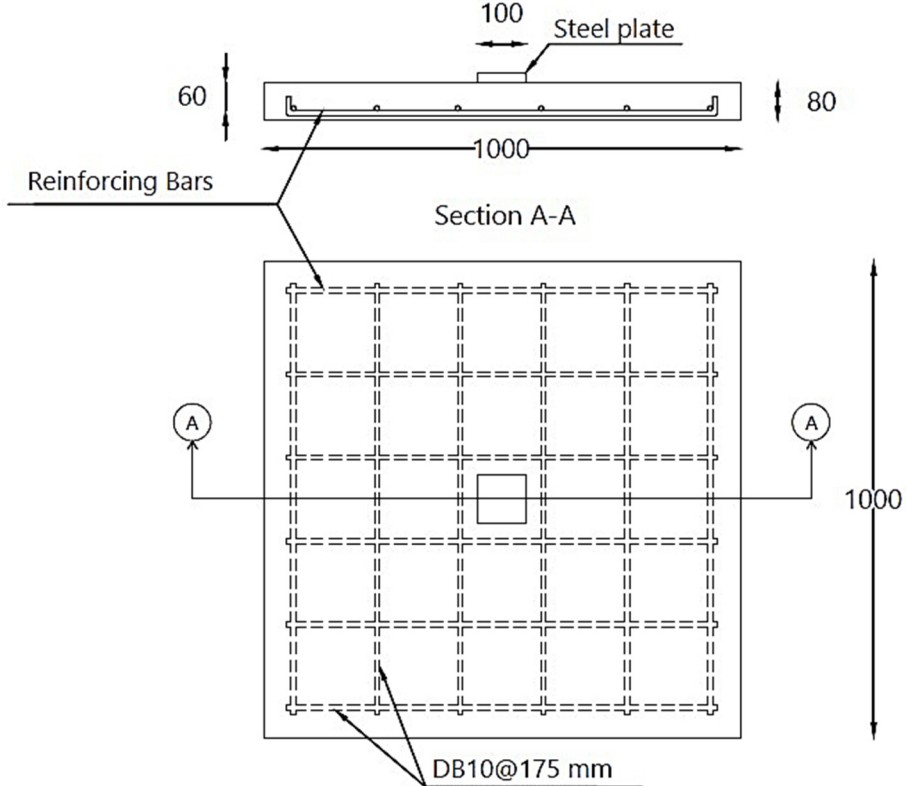

**Figure 3.** Typical layout of flexural reinforcement (units: mm).

As per the recommendations of ACI 318-19, critical shear perimeter $b_0$ was reduced to account for the presence of openings located within 4 times of slab's effective depth. Critical shear perimeter was reduced by an area enclosed by two tangents extended to the outline of openings from the center of column as shown in Figure 4. ACI 318-19 and Eurocode 2 consider the maximum location at which an opening can still affect the punching shear capacity of flat slabs to be 4H and 6d from column's face, respectively (where "H" and "d" are slab thickness and effective depth, respectively). It is to be mentioned that ACI 318-19 recently adopted this distance to be 4H. The previous edition, i.e., ACI 318-14 considered this distance to be 10H from column's face. This modification is based upon the findings of Genikomsou and Polak [20], who concluded that slabs with openings situated farther than 4H from column's face essentially exhibited same punching strength as those slabs without openings. Therefore, location of openings were well within the domain specified by ACI 318-19.

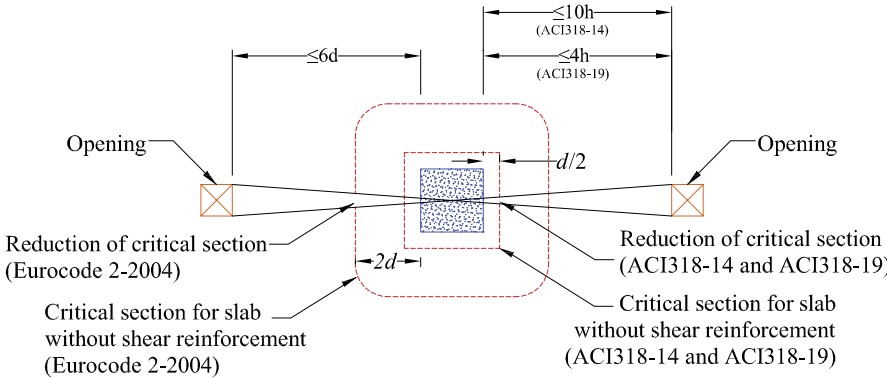

**Figure 4.** Calculation of $b_0$.

According to Eurocode 2 [19] the design punching shear resistance of a slab without punching shear reinforcement along the control section ($v_{Rd,c}$) may be calculated as shown in Equation (2)

$$v_c = C_{Rd,c}\, k(100\rho_1\, f_{ck}\,)^{\frac{1}{3}} \tag{2}$$

where

- $f_{ck}$ is in MPa
- $k = 1 + \sqrt{\frac{200}{d}} \le 2.0$ ($d$ in mm)
- $\rho_1 = \sqrt{\rho_{ly}\, \rho_{lz}} \le 0.02$
- $\rho_{ly}$, $\rho_{lz}$ related to the bonded tension steel in y- and z-direction respectively.
- $C_{Rd,c}$ is 0.18.

For Eurocode 2, design checks are carried out at the critical perimeter and at column's face. The code specifies this critical perimeter to be located at a distance of 2d from columns face (without shear reinforcement). Further, a part of this perimeter needs to be considered ineffective as long as the shortest distance between column's face and the opening does not exceed 6d. Size of ineffective perimeter is enclosed by two tangents ex-tending from the center of column to the outline of opening as illustrated in Figure 4.

Similarly, flexural strength of tested specimens was assessed as per yield line theory [21] descriptive equation as follows in Equation (3) and (4).

$$P_y = m_r \left( \frac{8}{\frac{L}{c} - 1} + 2\pi \right) \tag{3}$$

$$m_r = \rho d^2 f_y \left( 1 - 0.59\rho \frac{f_y}{f_c'} \right) \tag{4}$$

where

- $P_y$ = Flexural capacity
- $L$ = Length of supported slab
- $c$ = Loading plate side length
- $\rho$ = Flexural reinforcement ratio

Table 2 lists calculated flexural and shear strengths of all specimens. It can be seen that shear strengths of all specimens are significantly lower than their flexural strengths.

**Table 2.** Nominal flexural and punching shear capacities of tested specimens.

| Specimen No. | Specimen Name | Flexural Capacity (kN) | Shear Capacity (kN) | |
|---|---|---|---|---|
| | | | ACI 318-19 | Eurocode 2 |
| 1 | CON1 | 109.34 | 56.93 | 66.54 |
| 2 | 2S1H | 109.34 | 49.26 | 56.00 |
| 3 | 2C1H | 109.34 | 50.89 | 58.22 |
| 4 | 2R1H | 109.34 | 41.59 | 46.04 |
| 5 | 2S4H | 109.34 | 54.24 | 66.18 |
| 6 | 2C4H | 109.34 | 54.47 | 66.21 |
| 7 | 2R4H | 109.34 | 51.55 | 65.81 |
| 8 | CON2 | 115.15 | 69.07 | 75.70 |
| 9 | 4S1H | 115.15 | 50.47 | 51.72 |
| 10 | 4C1H | 115.15 | 54.44 | 50.20 |
| 11 | 4R1H | 115.15 | 31.89 | 29.07 |
| 12 | 4S4H | 115.15 | 62.94 | 74.86 |
| 13 | 4C4H | 115.15 | 63.11 | 74.94 |
| 14 | 4R4H | 115.15 | 56.03 | 74.02 |

### 2.3. Material Properties

Mix design of concrete included Portland type I cement with a maximum aggregate size of 9.5 mm. To test the compressive strength, 6 concrete cylinders (150 mm in diameter to 300 mm in height) were prepared. Cylinders were tested by the Compression Testing Machine at department of civil engineering, faculty of engineering, King Mongkut's University of Technology Thonburi, Bangkok to determine the compressive strength ($f_c'$) following the standard guides of ASTM standards ASTM C39/C39M [22]. The compressive strength of concrete was 20.18 MPa and 29.71 MPa for group 1 and 2, respectively. Longitudinal reinforcement comprised of 10 mm deformed steel bars having yield strength of 619 and 545 MPa for group 1 and 2, respectively. This could be due to the reason that steel bars were ordered from different suppliers at different times. Steel bars were spaced at 175 mm on center and placed in both orthogonal directions comprising a longitudinal reinforcement ratio of 0.6% in each direction.

### 2.4. Load Setup

Experimental setup consisted of a hydraulic jack (with capacity of 60 tons), steel frame, load cell, wide steel flanges, and concrete block as shown schematically in Figure 5 (see Figure 6 for actual test configuration). Wide steel flanges were used to support all sides of specimens. Hydraulic jack was used to apply monotonic load at the center of each specimen. The hydraulic jack was placed on the top of the load cell to distribute the load down until it broke through the surface of the specimen. Linear Variable Displacement Transducers (LVDTs) were used to capture vertical deflection of each specimen as well as to monitor the rotation in each specimen. Placement of LVDTs is illustrated in Figure 7. Body of LVDT was affixed to a magnetic stand while its core touch slab specimen.

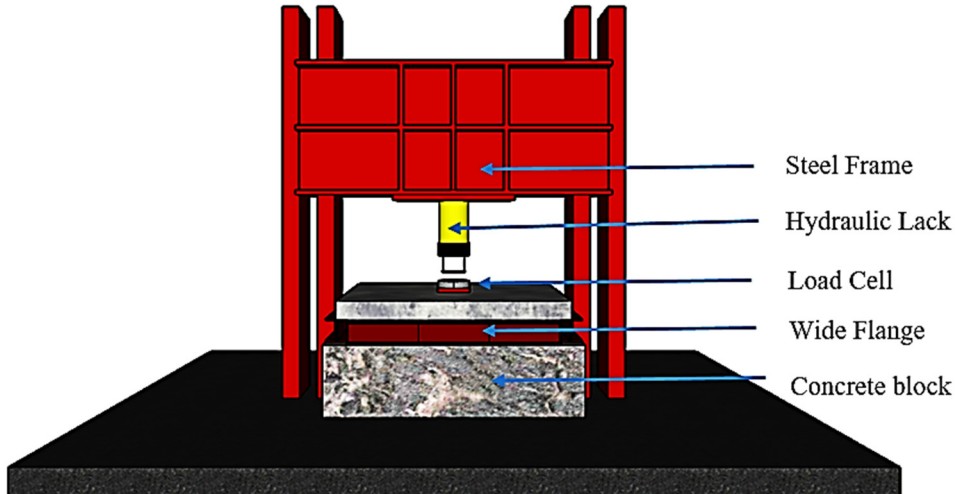

**Figure 5.** Load Setup.

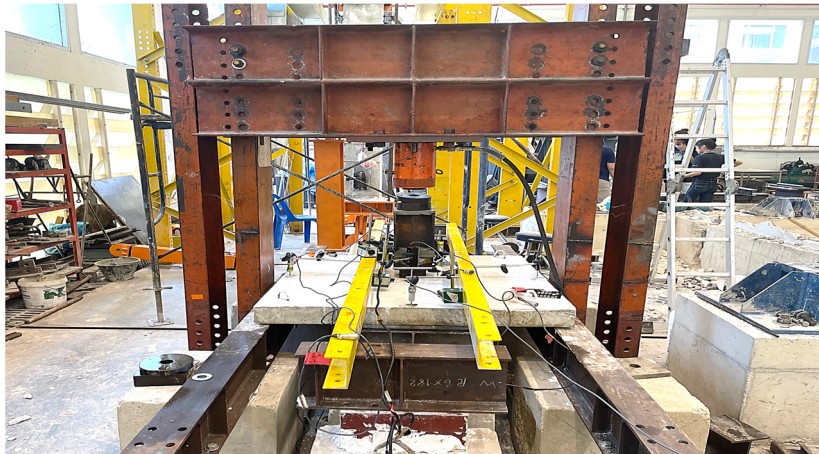

**Figure 6.** Actual test configuration.

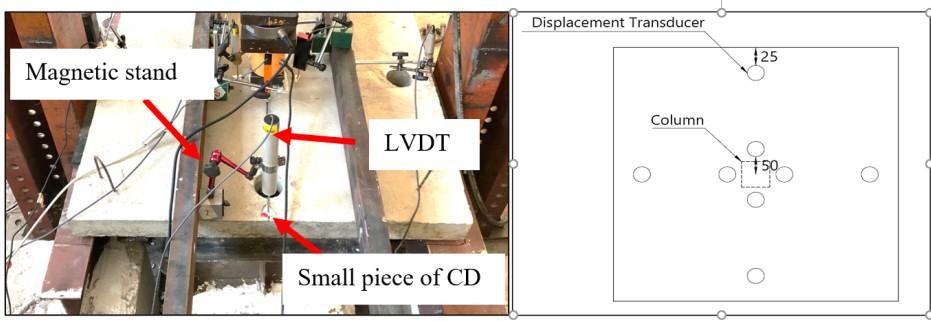

**Figure 7.** Location & placement of LVDTs.

## 3. Experimental Results

### 3.1. Failure Modes and Observations

Crack patterns of group 1 and 2 are presented in Figures 8–11. For the specimens in group 1, first cracks initiated from the face of columns. With further increase in load, cracks propagated towards the side of slab specimens. It can be seen in Figures 8 and 9 that the distance of openings from column's face had significant impact on the failure zone of specimens (dark solid lines). Failure zone was narrower and closer to the column when openings were located at a distance of 1 times of slab depth from the column face (Figures 8b–d and 9a). This signifies that relatively smaller slab area contributed towards

shear strength when openings were placed nearer the column. Further, for openings at 1H from the column, cracks also initiated from the openings that further deteriorated the shear capacity. On the contrary, specimens with openings located farther from column's face (i.e., at 4H from column's face) exhibited lesser cracking than their counterpart specimens and it is interesting to note that no crack initiated from these openings as well (Figures 8e,f and 9b). Specimens in group showed similar trend as those in group 1 as shown in Figures 10 and 11. Cracks intensity and propagation were more severe when openings were placed nearer to the column. Analogous to 4H openings in group 1, cracks did not initiate from openings at 4H from column's face in group 2 as well. A number of openings also had an adverse effect on shear resistant zone when increased from 2 to 4. For instance, specimen 2C1H had 2 circular openings at column's opposite side while 4C1H had four circular openings at each side of the column. Comparing their crack patterns at failure, it is interesting to observe that shear failure zone extended farther on sides where no openings were provided (case of 2C1H). Conversely, shear failure zone extended to equal distance from all four sides of column in case of 4C1H. This clearly explains the limit that openings impose on the shear capacity when placed nearer to the column's face. Comparing the crack patterns for different opening shapes, rectilinear openings allowed the initiation of cracks from all of their corners. Further, crack formation from rectilinear openings was denser as compared to circular section (Figures 10g and 11b).

ACI 318-19 recommends decreasing punching strength by reducing critical shear perimeter by an amount that is enclosed by straight lines starting from the centerline of column to the openings boundary as demonstrated in earlier sections. An important point is that this reduction should be made for an opening located at a distance smaller than 4H from column's face, where "H" is slab thickness. This highlights that an opening located farther than 4H from column's face does not affect the punching capacity. This explanation can be related to the crack patterns obtained in this study. For openings located at 1H from column's face, primary cracks (bold black lines) were located at the same location indicating that only a reduced slab area around column resisted punching shear. Shifting the openings away from column's face resulted in a wider periphery of primary cracks emphasizing on the availability of even larger slab area to resist punching shear. Another observation made is that placing openings on only two sides of columns resulted in a wider extension of primary cracks on sides without openings (for instance, see crack pattern of 2S1H). This states that the sides without openings contribute more towards resisting punching shear than the sides with openings. This transformation in the behavior of slabs from bi-directional to unidirectional action has been reported [16]. In addition, in all slab, severe cracks were also observed on the compression side along with the punching of the loading plate into the slab as shown in Figure 12.

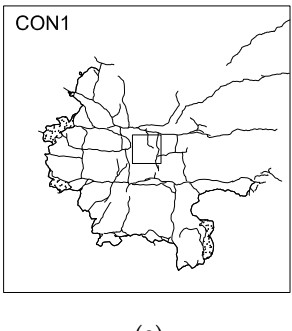

(**a**)

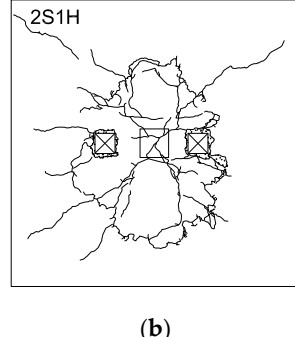

(**b**)

**Figure 8.** *Cont.*

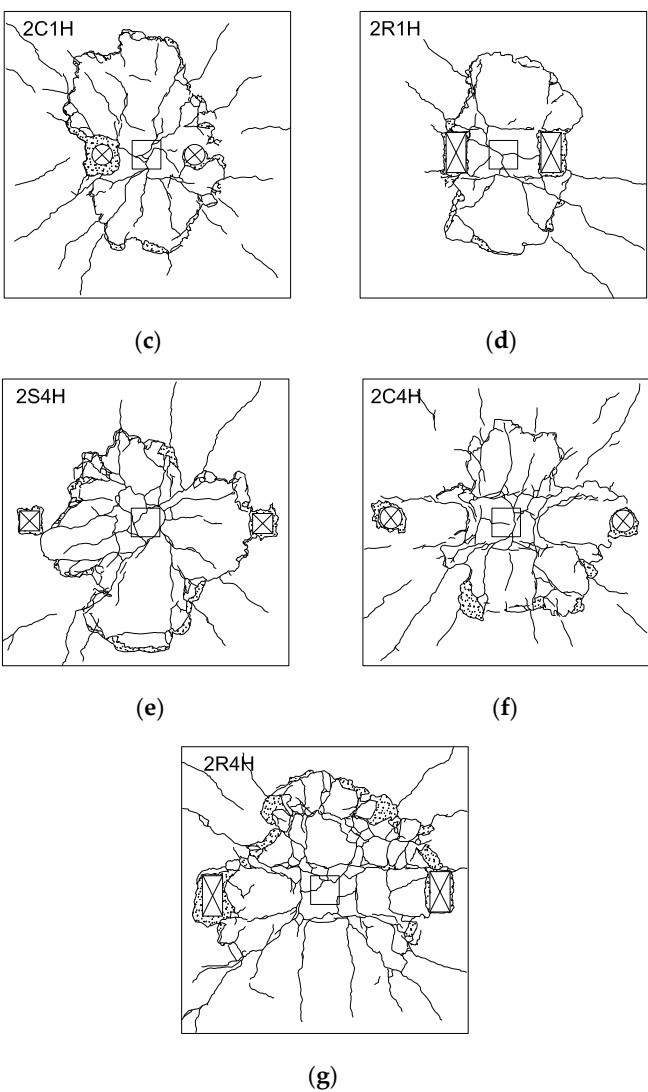

**Figure 8.** Crack patterns at failure for group 1 (Line diagrams). (**a**) Specimen CON1; (**b**) Specimen 2S1H; (**c**) Specimen 2C1H; (**d**) Specimen 2R1H; (**e**) Specimen 2S4H; (**f**) Specimen 2C4H; (**g**) Specimen 2R4H.

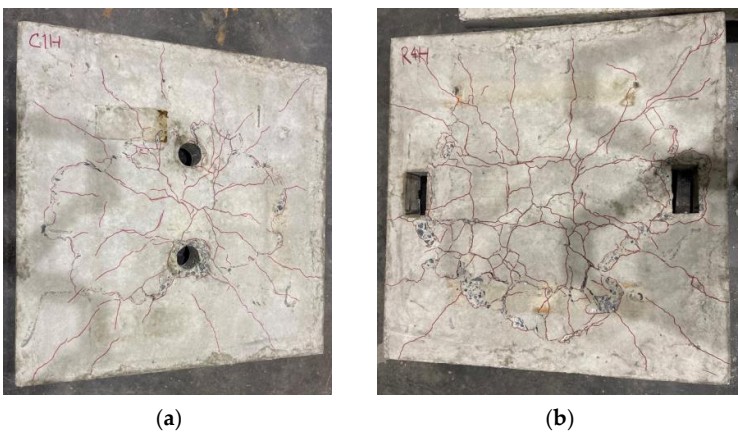

**Figure 9.** Crack patterns at failure for group 1 (Actual photos). (**a**) Specimen 2C1H; (**b**) Specimen 2R4H.

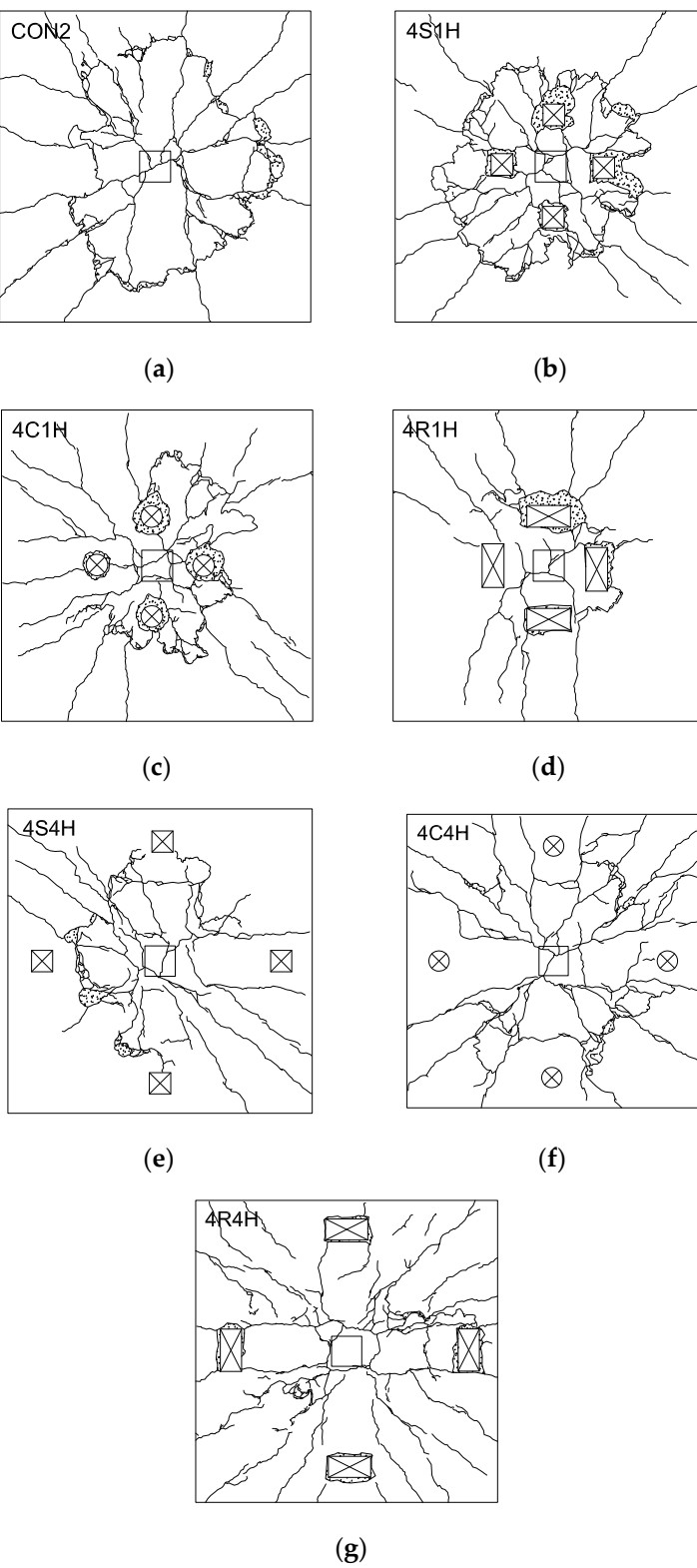

**Figure 10.** Crack patterns at failure for group 2 (Line diagrams). (**a**) Specimen CON2; (**b**) Specimen 4S1H; (**c**) Specimen 4C1H; (**d**) Specimen 4R1H; (**e**) Specimen 4S4H; (**f**) Specimen 4C4H; (**g**) Specimen 4R4H.

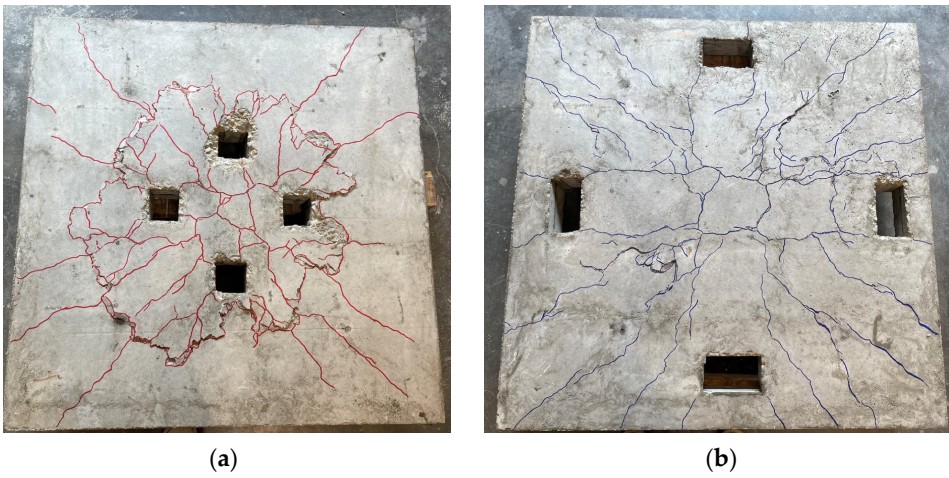

| (a) | (b) |

**Figure 11.** Crack patterns at failure for group 2 (Actual photos). (**a**) Specimen 4S1H; (**b**) Specimen 4R4H.

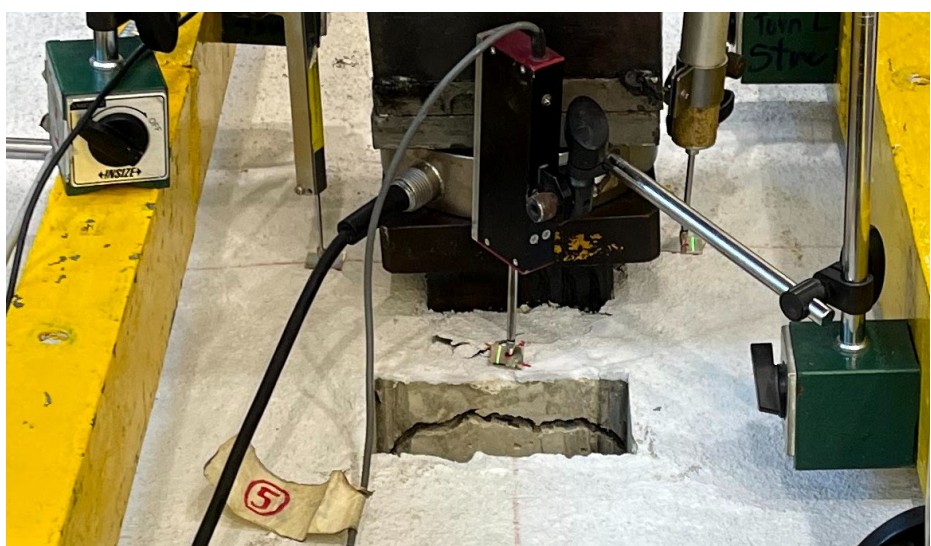

**Figure 12.** Typical cracks and punching of steel plate on compression side.

### 3.2. Ultimate Rotation

Rotation of each slab at failure was approximated by using the LVDT values. Figure 13 shows the ultimate rotation ($O_u$) of each specimen or the average angle in each specimen at failure. It can be seen that the ultimate rotations of slabs when openings were located at 4-time thickness of the slab in both groups is very close to that of the control specimen. However, reducing the distance to 1H from column's face drastically affected the ultimate rotation capacity of the specimens. Further, ultimate rotation of slabs with circular openings were highest (among slab with openings), followed by their counterpart slabs with square and rectangular openings, respectively. This trend is more dominant in the slabs with two openings.

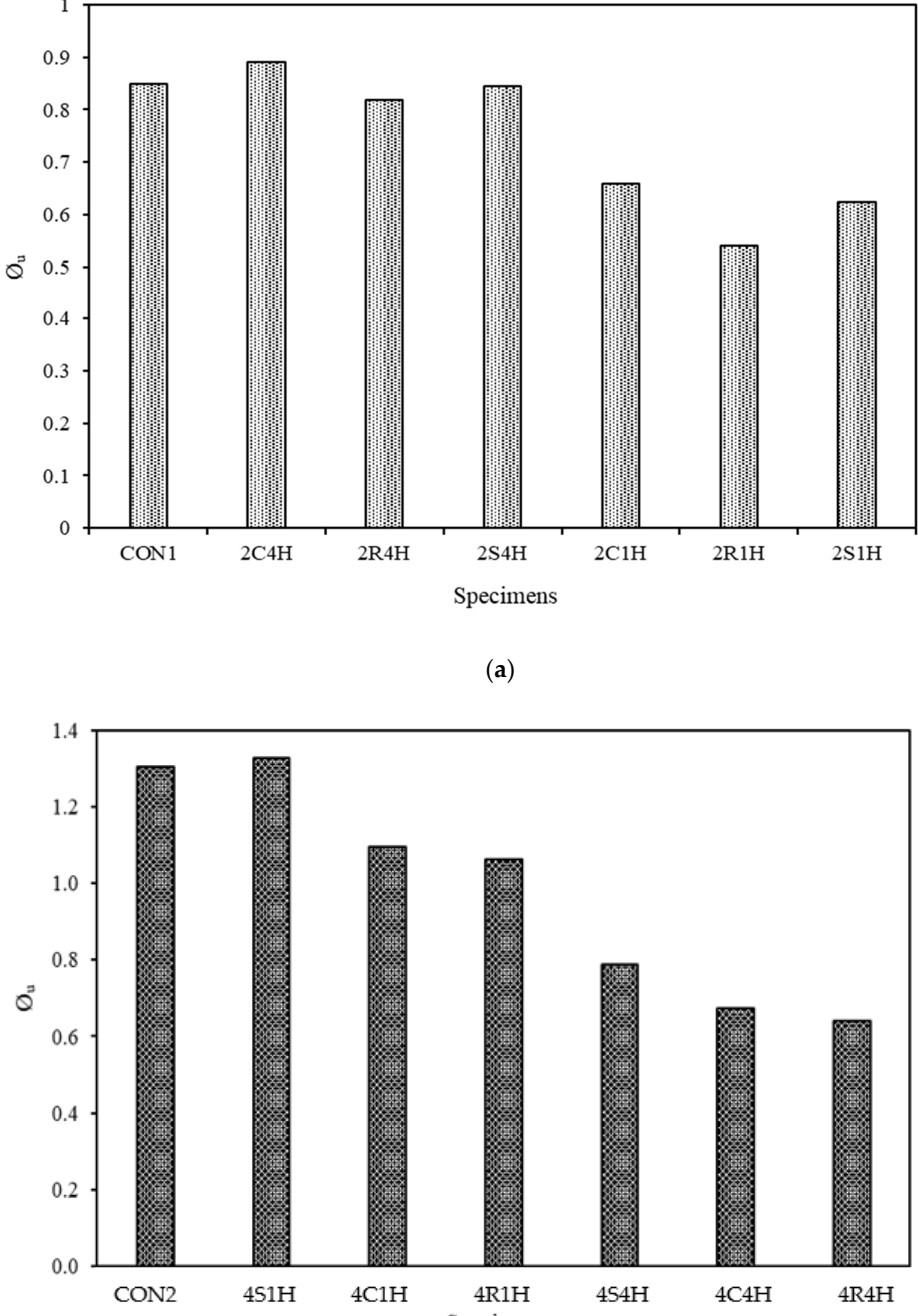

**Figure 13.** Ultimate rotations of slabs in (**a**) 2 openings (**b**) 4 openings.

### 3.3. Load vs. Deflection

Experimental results in terms of peak loads corresponding to punching shear for each specimen along with their corresponding maximum deflections are listed in Table 3. It is to be mentioned that the tested 2 groups of specimens had different concrete strengths since both concrete groups belonged to different batches ordered at different times. To allow for the comparison between two groups, experimental strength was normalized to theoretical strength $0.33 \sqrt{f'_c} b_o d$.

**Table 3.** Summary of test results.

| Specimen No. | Specimen Name | Compressive Strength of Concrete (MPa) | Peak Load (kN) | Peak Deflection (mm) | Failure Mode |
|---|---|---|---|---|---|
| 1 | CON1 | 20.18 | 57.73 | 7.745 | Punching Shear |
| 2 | 2S1H | 20.18 | 49.09 | 5.479 | Punching Shear |
| 3 | 2C1H | 20.18 | 51.10 | 5.835 | Punching Shear |
| 4 | 2R1H | 20.18 | 44.71 | 4.845 | Punching Shear |
| 5 | 2S4H | 20.18 | 57.04 | 7.616 | Punching Shear |
| 6 | 2C4H | 20.18 | 56.92 | 7.802 | Punching Shear |
| 7 | 2R4H | 20.18 | 55.50 | 7.236 | Punching Shear |
| 8 | CON2 | 29.71 | 70.15 | 11.829 | Punching Shear |
| 9 | 4S1H | 29.71 | 45.84 | 5.677 | Punching Shear |
| 10 | 4C1H | 29.71 | 49.06 | 7.244 | Punching Shear |
| 11 | 4R1H | 29.71 | 38.89 | 6.029 | Punching Shear |
| 12 | 4S4H | 29.71 | 67.61 | 9.299 | Punching Shear |
| 13 | 4C4H | 29.71 | 66.34 | 12.195 | Punching Shear |
| 14 | 4R4H | 29.71 | 63.85 | 9.847 | Punching Shear |

### 3.3.1. Location of Openings

Figure 14 illustrates the comparison of load vs. deflection curves of specimens with openings at 4H and 1H from column's face, respectively. As evident from Figure 14b, placing openings near the column's face (at 1H) irrespective of their shape had an adverse effect on the punching shear capacity of the slabs. When the distance of openings was increased from 1H to 4H (where H is slab thickness), peak loads sustained by each shape of openings were found to be very close to the peak load of the control specimen (see Figure 14a). This emphasizes the risk of placing openings nearer than 4H from the face of column. The reason can be attributed to the decreased critical perimeter when the openings were placed at 1H that substantially reduced shear capacities of respective slabs. Specimens in group 2 (see Table 3) exhibited similar response as those in group 1. Placing 4 openings at 1H deteriorated shear capacity more severely than 4 openings at 4H from column's face. These results agree well with the previous findings [9,11].

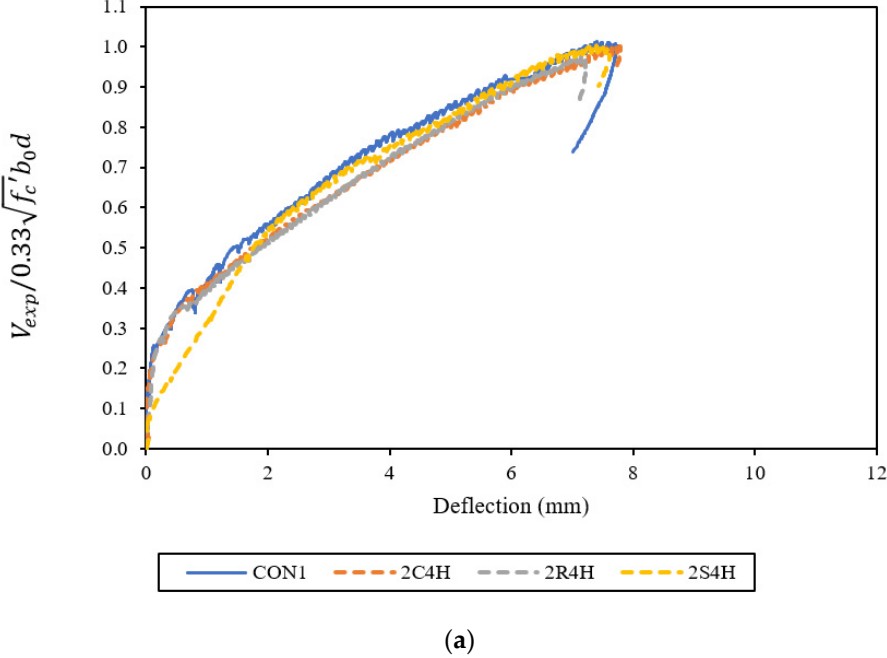

(**a**)

**Figure 14.** *Cont*.

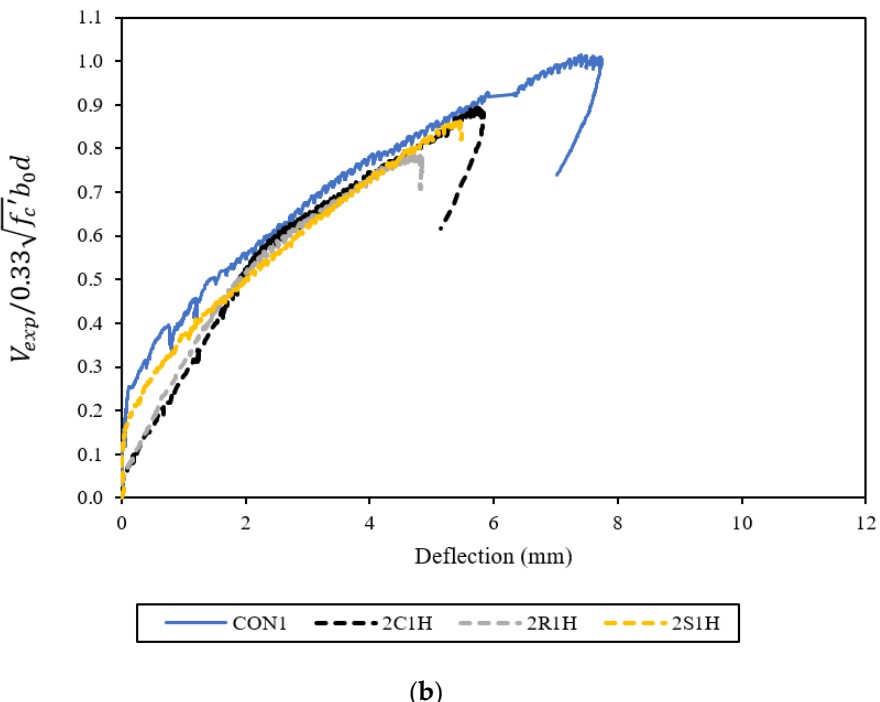

**(b)**

**Figure 14.** Normalized load-displacement curves (**a**) Openings at 4H (**b**) Openings at 1H.

3.3.2. Shape of Openings

Specimens numbered from 1 to 7 belong to the 1st group having 2 openings while the rest of the specimens belong to group 2 each having 4 openings. The control specimen in group 1 (CON1) was able to sustain highest load i.e., 57.73 kN. For the same distance from column's face (group 1), specimens with circular openings were able to resist highest loads followed by square and rectangular openings, respectively. For instance, peak load resisted by specimen 2C1H was 51.10 kN whereas corresponding values for specimens 2S1H and 2R1H were 49.09 and 44.71 kN, respectively. A similar trend was observed in peak deflection values as circular specimen underwent highest deflections followed by square and rectangular specimens, respectively. As illustrated in Figure 15, slabs with circular openings outperformed slabs with square and rectangular openings. These findings agree with the previous findings [17] where circular openings had a lower deteriorating effect on punching capacities than rectangular openings placed at the same location in flat slabs. There can be two reasons behind this. Firstly, rectangular openings had the largest area followed by square and circular openings, respectively. Secondly, it has been known that stress concentrations at the corners of rectilinear sections adversely affect their performance in comparison to circular shape where a uniform stress distribution can be assumed. For the openings located at 4H from the column's face, peak loads sustained by circular and square openings were comparable. However, rectangular openings still created the lower bound in terms of peak load. This can be attributed to larger areas of rectangular openings in comparison with circular and square shape openings.

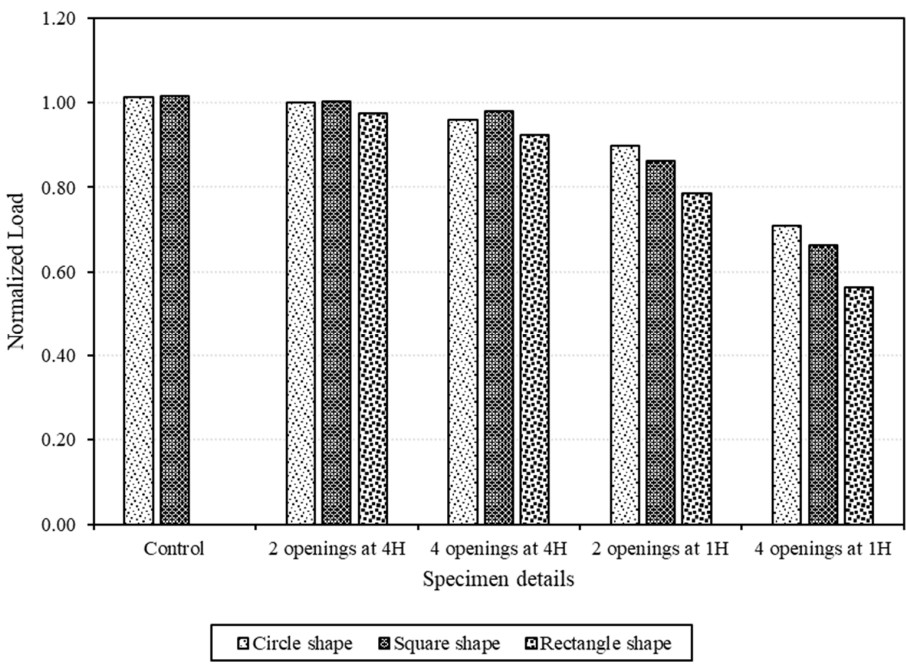

**Figure 15.** Comparison between square, circle, and rectangle shape.

### 3.3.3. Number of Openings

The number of openings also had an adverse impact upon the shear capacity. For comparison, reduction in peak loads (at 1H) for circular, square, and rectangular openings in group 1 were 11.48, 14.97, and 22.44% respectively (see Figure 16a). Peak loads for the same specimens but with 4 circular, square, and rectangular openings (at 1H) were reduced by 30.06, 36.65, and 44.56%, respectively and as shown in Figure 16b.

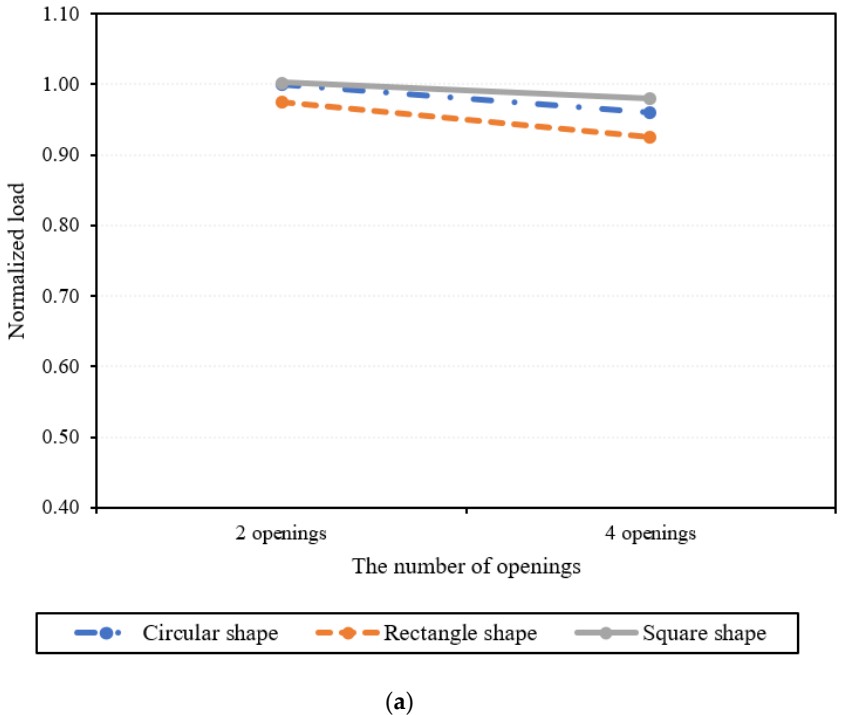

(a)

**Figure 16.** *Cont.*

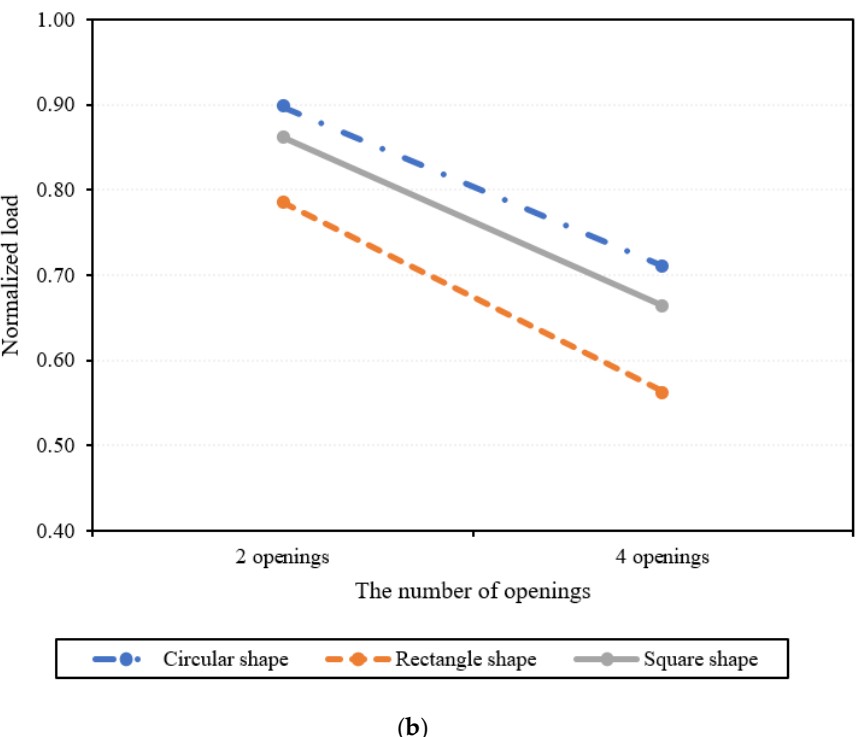

(**b**)

**Figure 16.** Punching shear capacity of the different the number of openings (**a**) Openings at 4H (**b**) Openings at 1H.

## 4. Comparison between Experimental Results & Analytical Predictions

The two approaches used to predict the punching shear capacity of tested specimens are based upon the recommendations of ACI 318-19 and Eurocode 2. Detailed explanations of the calculation procedures are already presented in previous sections. Here, first the major differences between these two codes are presented. ACI 318-19 states that the critical perimeter for punching shear is located at a distance of d/2 from column's face (Figure 17) while the same is located at 2d from column's face as per Eurocode 2 recommendations (where d is the effective depth of the slab) as shown in Figure 18. Further, an opening can have significant effect on punching capacity up to a distance of 4H and 6d (where "H" and "d" are slab thickness and effective depth, respectively) from column's face defined by ACI 318-19 and Eurocode 2, respectively. Distances of openings were chosen to be within the domain specified by these codes.

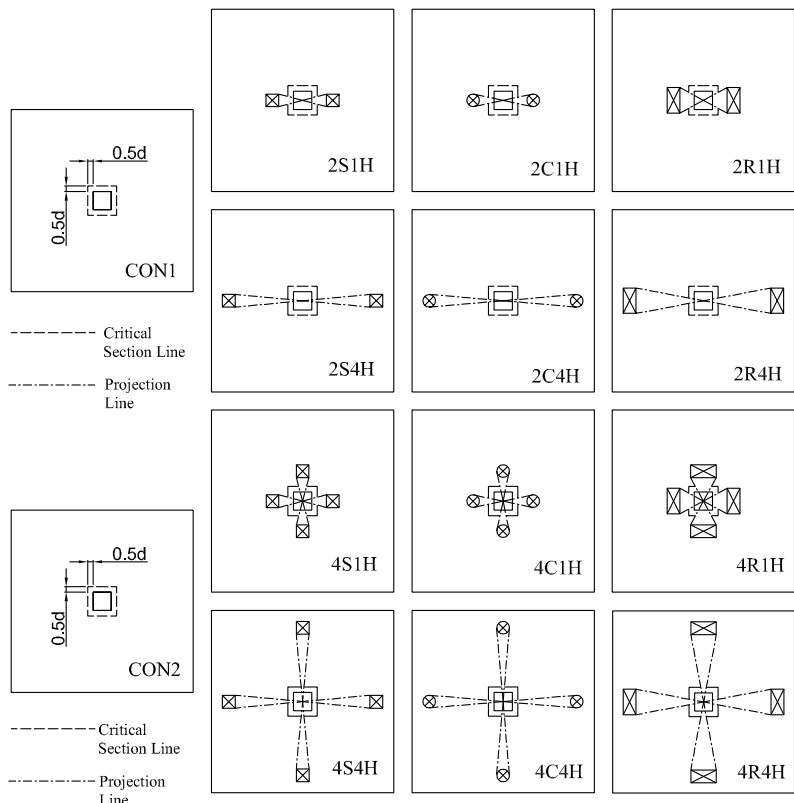

**Figure 17.** Definition of critical perimeter as per ACI 318-19.

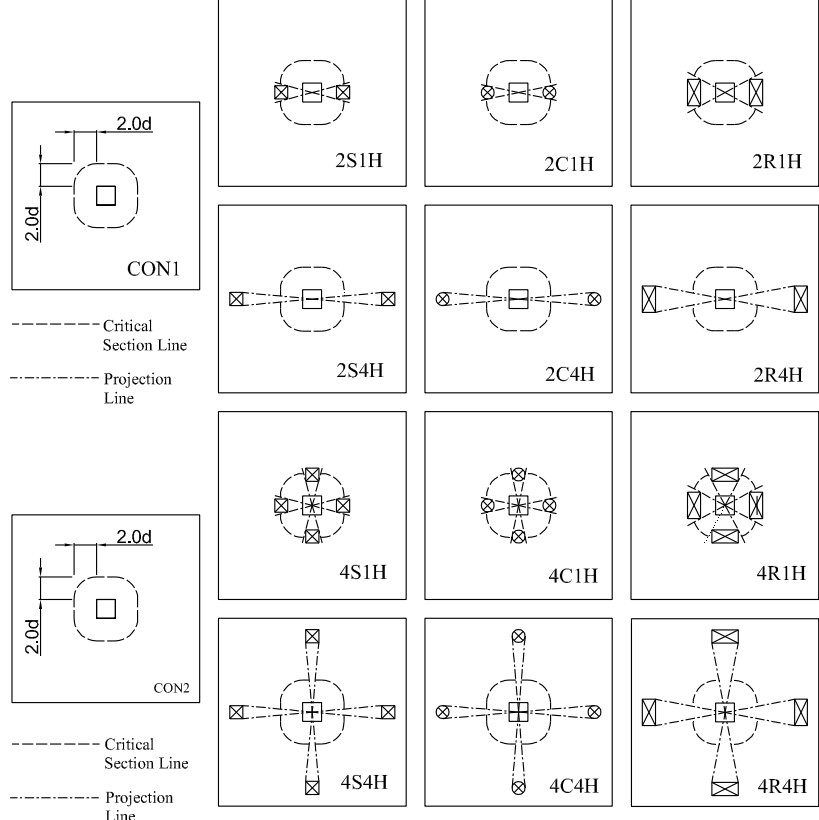

**Figure 18.** Definition of critical perimeter as per Eurocode 2.

Table 4 presents a comparison between experimental and analytical predicted punching shear capacities of all specimens. It was found that the average difference be-tween experimental and analytical predictions of ACI 318-19 was 4% while the same for Eurocode 2 was 7%. Figure 19 presents the variation of theoretical predictions from experimental results. Punching strengths of control specimens CON1 and CON2 predicted by ACI 318-19 were offset only by −1 and −2%, respectively, from their corresponding experimental values. The same had differences of +13 and +7%, respectively, for Eurocode 2 predictions. ACI 318-19 predictions for control specimens (without openings) were not only closer to the experimental results than Eurocode 2 predictions but they also provided more conservative results. Among group 1 specimens, the largest percentage difference obtained in ACI 318-19 predicted strengths was −8%. The counterpart value for Eurocode 2 results was +16%. It was also observed that ACI 318-19 consistently predicted conservative strengths among group 1 specimens. On the contrary, Eurocode 2 constantly overestimated punching capacities of all the specimens in group 1. Apart from specimens 4S1H and 4C1H in group 2, ACI 318-19 predictions were conservative for the remaining specimens while Eurocode 2 again overestimated punching strengths of all specimens but 4R1H. It is to be mentioned that the error obtained in theoretical predictions of both codes varied with the shape of openings irrespective of their number and location. Therefore, this greatly emphasizes on the need of further research to account for the shape of openings in determining their deteriorating effect on punch capacities of flat slabs. Statistically, ACI 318-19 recommendations yielded a slightly lower standard deviation from experimental results than Eurocode 2. Further, the mean of the ratio of experimental to analytical results was 1.04 and 0.93 for ACI 318-19 and Eurocode 2, respectively.

**Table 4.** Comparison between experimental and analytical punching shear strengths.

| No. | Name | $f'_c$ (MPa) | Experimental (kN) | Reduction (%) | Analytical Punching Capacity (kN) | | Ratio of Experimental to Analytical | |
|---|---|---|---|---|---|---|---|---|
| | | | | | ACI318 | Eurocode 2 | ACI318 | Eurocode 2 |
| 1 | CON1 | 20.18 | 57.73 | - | 56.93 | 66.54 | 1.01 | 0.87 |
| 2 | 2S1H | 20.18 | 49.09 | 14.97 | 49.26 | 56.00 | 1.00 | 0.88 |
| 3 | 2C1H | 20.18 | 51.10 | 11.48 | 50.89 | 58.22 | 1.00 | 0.88 |
| 4 | 2R1H | 20.18 | 44.71 | 22.55 | 41.59 | 46.04 | 1.08 | 0.97 |
| 5 | 2S4H | 20.18 | 57.04 | 1.20 | 54.24 | 66.18 | 1.05 | 0.86 |
| 6 | 2C4H | 20.18 | 56.92 | 1.40 | 54.47 | 66.21 | 1.04 | 0.86 |
| 7 | 2R4H | 20.18 | 55.50 | 3.86 | 51.55 | 65.81 | 1.08 | 0.84 |
| 8 | CON2 | 29.71 | 70.15 | - | 69.07 | 75.70 | 1.02 | 0.93 |
| 9 | 4S1H | 29.71 | 45.84 | 34.65 | 50.47 | 51.72 | 0.91 | 0.89 |
| 10 | 4C1H | 29.71 | 49.06 | 30.06 | 54.44 | 50.20 | 0.90 | 0.98 |
| 11 | 4R1H | 29.71 | 38.89 | 44.56 | 31.86 | 29.07 | 1.22 | 1.34 |
| 12 | 4S4H | 29.71 | 67.61 | 3.62 | 62.94 | 74.86 | 1.07 | 0.9 |
| 13 | 4C4H | 29.71 | 66.34 | 5.43 | 63.11 | 74.94 | 1.05 | 0.89 |
| 14 | 4R4H | 29.71 | 63.85 | 8.98 | 56.03 | 74.02 | 1.14 | 0.86 |
| | | | | Average | | | 1.04 | 0.93 |
| | | | | Standard Deviation (SD) | | | 0.08 | 0.12 |

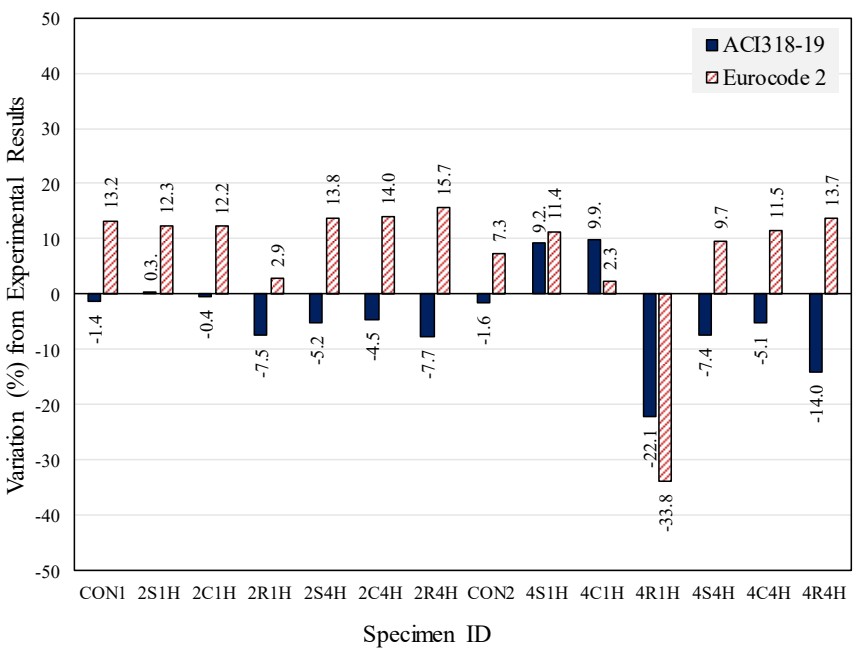

**Figure 19.** Variation (%) of theoretical predictions from experimental results.

## 5. Discussion

This study investigated into the effects of openings on punching shear capacity of flat slabs. For this purpose, 14 flat slab specimens were tested in 2 groups depending upon the concrete strength. The novelty of this research lies in the adoption of different shapes of openings for investigating their effects on punching capacities of flat slabs. Although, it has been known from literature and current design codes (see earlier sections) that the distance of an opening has inverse relation with its deteriorating effect on punching capacity. However, this effect has not been related with the shape of the opening in the past. This study considers all possible factors (including distance from column's face, shape, and their number) associated to the openings around columns in flat slabs to study their effects on punching capacities.

It is found that the shape of opening actually affects punching capacity of flat slabs. Circular, square, and rectangular openings were used in this study. For the same distance from column's face, circular openings are found to have least effect on punching strength, followed by square and rectangular openings, respectively. Stress concentrations at the corners of square and rectangular openings can be the catalyst for this excessive degradation of punching strength. Further, circular shape have smaller area than square opening for the same diameter and side dimension. The second factor related to openings was their number. In this study, 2 and 4 openings were used around column's periphery. It was expected from the slabs with 4 openings to have smaller punching strength than the slabs with 2 openings. Results have confirmed this as in both the groups of test matrix, increasing the openings adversely affected the punching strength. The last parameter considered was the distance of openings from the column's face. Openings were placed at 1 and 4 times the slab thickness "H" from column's face. Results suggest that openings do not have significant effect on punching strength when placed at 4H from column's face. However, a substantial decrease in punching capacity was observed for slabs with openings located at 1H from column's face.

Code provisions of ACI 318-19 and Eurocode 2 were deployed to predict the punching capacities of flat slabs. It is found that ACI 318-19 slightly edges past Eurocode 2 in predicting punching capacities of flat slabs with openings. Nevertheless, ACI 318-19 predictions were constantly conservative while Eurocode 2 overestimated punching strengths. ACI 318-19 states that the punching strength of flat slabs must be decreased by an amount that is proportional to the reduced critical perimeter. The amount by which the critical

perimeter must be reduced has already been stated in earlier sections. An important result in this regard is that ACI 318-19 updated its clause about the distance from column's face within which an opening will have adverse effect on punching strength. This distance was updated to 4H that previously was 10H in ACI 318-14. Findings of this research agree with the recent adoption of 4H by ACI 318-19. Openings, irrespective of their shape and number, had a negligible effect on punching capacity when they were placed at 4H from column's face. An important consideration in this regard is the shape of openings. Openings of same number and at same distance from column's face but with different shapes resulted in different punching shear strengths. Though code provisions of ACI 318-19 and Eurocode 2 account for this effect by reducing critical perimeter by an amount that is a function of the size of an opening, their predictions for different opening shapes resulted in large scatter. The predictions of both codes for square and circular openings (same number of openings and at same distance from column's face) resulted in a close variation from experimental results. For example, Eurocode 2 predictions for 2S1H and 2C1H (i.e., 2 square and circular openings at 1H from column's face, respectively) varied from experimental results by +12%. But it resulted in a variation of +2.9% for the same number and distance of rectangular openings. A similar trend could also be observed in ACI 318-19 predictions (see Figure 19). Therefore, it is much desired to extend this area of research to extend the knowledge about the effects of openings on punching capacities with special attention to their shapes.

## 6. Conclusions

An experimental study was conducted to study the effect of different shapes of openings in flat slabs at different distances from a concentrically loaded column's face. A total of 14 specimens were tested in 2 groups each consisting of 7 specimens. Group 1 specimens were provided with 2 openings while 4 openings were provided in group 2 specimens. Effects of shape (circular, square, and rectangular), number (2 and 4), and distance of openings from column's face (1H and 4H where H is slab's thickness) on punching shear capacity were carefully investigated. Key conclusions drawn from this study are summarized in this section.

1.  In terms of the shape of openings, punching capacity of flat slabs were least affected by circular openings followed by square and rectangular openings, respectively.
2.  Openings provided at 1H from column's face drastically reduced the punching shear capacity of flat slabs. On the contrary, openings provided at 4H from column's face had little impact on slab's shear capacity. Rectangular openings had the worst effect on punching capacity for all opening shapes at 4H from column's face. Rectangular openings at 4H reduced punching capacity by 3.86 and 8.98% for 2 and 4 openings, respectively. Reduction caused by circular and square and circular openings was comparable at 4H from column's face and irrespective of their number.
3.  Irrespective of the shape of the openings, increasing the number of openings from 2 to 4 substantially reduced the punching capacity of slabs.
4.  Ultimate rotation of slabs with openings at 1H from column's face were significantly lower than their counterpart slabs with openings at 4H from column's face.
5.  Punching shear capacities were analytically predicted following the descriptive equations of ACI 318-19 and Eurocode 2. Ratio of experimental to analytical predictions of ACI 318-19 and Eurocode 2 were 1.04 and 0.93 suggesting that ACI 318-19 recommendations are slightly closer to the real case scenarios. This reason can be attributed to the different locations of critical punching shear perimeter from column's face as stated by the two codes.
6.  ACI 318-19 consistently predicted conservative and more closer punching capacities than those predicted by Eurocode 2. For openings located at same distance and of same number, the variation of code predictions from experimental results was marginal for square and circular openings. For instance, ACI 318-19 predicted punching strengths for specimens 2S4H and 2C4H to be −5 and −4% lower than their respective experimental values. However, this was not true for rectangular openings

as this error increased to −8% for the specimen 2R4H. Similarly, the predictions of ACI 318-19 for 2S1H and 2C1H resulted in a variation of about −0.4% from their experimental results. Corresponding prediction for the rectangular openings (i.e., for specimen 2R1H) resulted in a variation of −7.5%. A similar trend in the predictions of Eurocode 2 was also observed. Therefore, further research is desired in area to account for the shape of openings in determining their deteriorating effect on punching capacities of flat slabs.

**Author Contributions:** Conceptualization, E.Y., Y.T., T.W. and J.A.; methodology, E.Y., Y.T., T.W. and J.A.; investigation, E.Y., Y.T., T.W. and J.A.; writing—original draft preparation, E.Y., Y.T., T.W., J.A., P.J. and Q.H.; writing—review and editing, E.Y., Y.T., T.W., J.A., P.J. and Q.H. All authors have read and agreed to the published version of the manuscript.

**Funding:** This work was financially supported by Office of the Permanent Secretary, Ministry of Higher Education, Science, Research and Innovation under Grant No. RGNS 63-093.

**Institutional Review Board Statement:** Not applicable.

**Informed Consent Statement:** Not applicable.

**Acknowledgments:** The authors also thank the staff of CIVIL-KMUTT for their technical support in the laboratory.

**Conflicts of Interest:** The authors declare no conflict of interest.

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
