# Peer review of "Effect of Shape, Number, and Location of Openings on Punching Shear Capacity of Flat Slabs"

_buildings, doi:10.3390/buildings11100484_

Round 1

Reviewer 1 Report

The topic of the paper is interesting, only some parts need to be specified in detail or corrected.

  • Line 74 – what was the reason for the distance of opening position 4*h? Is it relevant? Smaller distances should have a more informative value.
  • Line 131 – add units
  • Line 153 – authors write about “standard concrete cylinders” – it is needed to name the standard or the dimensions.
  • Line 157 – what is the reason for so different vales of yield strengths? Explain.
  • Line 158 – it would be better to mention the reinforcement ratio.
  • Line 189 – these results are well known and expected because of the position of the openings.
  • Lines 293-294 – this conclusion is only valid for openings in 1H (as seen from Fig. 15)
  • Line 379 – some of the conclusions do not bring new information, these facts are well known are well expected.

Reviewer 2 Report

The paper deals with an experimental campaign performed to evaluate and analyse the reduction of the punching shear capacity of flat slabs with the distance of openings from the columns. In particular, 7 experimental results of 14 flat specimens have been investigate, that change the number, the shape 8 and the location of openings. Then, the values of the punching shear capacity have been compared with that obtained by 14 the descriptive equations of ACI 318-19 and Eurocode 2.

In the reviewer opinion, the paper handles an interesting and of great importance issue for the design of structure. Nevertheless, the results seem to confirms the design prescription provided by the analysed codes, so the originality and the motivation of the paper should be described with more emphasis.

Moreover, in the reviewer opinion:

  • rows 116-119: this affirmation on the difference between the ACI 318-14 and ACI 318-19 should be better motivated (or eliminated, it doesn’t provide to the paper)
  • rows 255-258: motivate, compare and explain the results obtained with the 2 strength concrete classes
  • Figure 14a e 14b should be improved by employing the same scale on the abscissa axis
  • Figura 16a e 16b should be improved by employing the same scale on the ordinate axis
  • A more detailed discussion should be inserted on the comparison with the analytical results, highlighting the difference between the different formulation and results. Further, some discussion on the best formulation is missing, and further any possible consideration on how to improve the actual code’s formulations.

After satisfied these required reviews, in the reviewer opinion the paper could be accepted for publication.

Reviewer 3 Report

The author(s) described the experimental programme on the influence of opening number, shape and position on the punching shear strength of flat rc slabs. The experimental results are compared to the values obtained by the building code design formulas. The study revealed the weaknesses of rc flat slabs with respect to observed opening cases, as well as the mismatch with the building codes formulas. 

The manuscript could be improved in the 1. Introduction and 5. Discussion section. 

The following comments and suggestions should be considered before the publication:

  • in 1. Introduction section:
  • lines 27-30, please elaborate the main findings (and cases considered) in references [6-16]
  •  please elaborate in brief the current building code requirements on the punching shear capacity by taking the near openings into account
  • In 5. Discussion section:
  • please add an overall graphical representation of Table 4, and please point out the cases where the building codes underestimated the punching shear capacity (state possible explanations) 
  • please add a brief comparison with the previous experimental studies mentioned in introduction

The author(s) are complimented for their work. 

Round 2

Reviewer 3 Report

The author(s) have addressed the reviewer's comments and suggestions. 
